# Endometriosis and the Fallopian Tubes: Theories of Origin and Clinical Implications

**DOI:** 10.3390/jcm9061905

**Published:** 2020-06-18

**Authors:** Christopher J. Hill, Marwa Fakhreldin, Alison Maclean, Lucy Dobson, Lewis Nancarrow, Alice Bradfield, Fiona Choi, Diandra Daley, Nicola Tempest, Dharani K. Hapangama

**Affiliations:** 1Centre for Women’s Health Research, Department of Women’s and Children’s Health, Institute of Life Course and Medical Sciences, University of Liverpool, Member of Liverpool Health Partners, Liverpool L8 7SS, UK; C.J.Hill1@liv.ac.uk (C.J.H.); Alison.Maclean@liverpool.ac.uk (A.M.); Lucy.Dobson@liverpool.ac.uk (L.D.); L.Nancarrow@liverpool.ac.uk (L.N.); A.J.Bradfield@liverpool.ac.uk (A.B.); F.Choi2@liverpool.ac.uk (F.C.); diandra.daley@nhs.net (D.D.); Nicola.Tempest@liverpool.ac.uk (N.T.); 2Liverpool Women’s NHS Foundation Trust, Member of Liverpool Health Partners, Liverpool L8 7SS, UK; marwa.fakhreldin@lwh.nhs.uk

**Keywords:** endometriosis, fallopian tubes, pathogenesis, stem cells, ovarian cancer

## Abstract

Endometriosis is a common, oestrogen driven chronic condition, where endometrium-like epithelial and stromal cells exist in ectopic sites. At present, no curative treatments are available and the existing evidence for disease progression is conflicting. The pathogenesis is still unknown and evidently complex, as mechanisms of initiation may depend on the anatomical distribution of endometriotic lesions. However, amongst the numerous theories and plethora of mechanisms, contributions of the fallopian tubes (FT) to endometriosis are rarely discussed. The FT are implicated in all endometriosis associated symptomatology and clinical consequences; they may contribute to the origin of endometriotic tissue, determine the sites for ectopic lesion establishment and act as conduits for the spread of proinflammatory media. Here, we examine the available evidence for the contribution of the human FT to the origin, pathogenesis and symptoms/clinical consequences of endometriosis. We also examine the broader topic linking endometriosis and the FT epithelium to the genesis of ovarian epithelial cancers. Further studies elucidating the distinct functional and phenotypical characteristics of FT mucosa may allow the development of novel treatment strategies for endometriosis that are potentially curative.

## 1. Introduction

Endometriosis is a common, oestrogen driven chronic condition, where endometrium-like epithelial and stromal cells exist in ectopic sites beyond their native location; the internal lining of the uterine cavity. Approximately 10% of reproductive age women are affected by endometriosis, which equates to around 190 million women worldwide [1,2]. It is most commonly associated with chronic pelvic pain (CPP) and accounts for up to 70% of such diagnoses [3]. Other associated symptoms include pain during periods (dysmenorrhoea) and/or sexual intercourse (dyspareunia), intermenstrual bleeding and infertility [4]. The most common sites for endometriotic lesions to establish are the ovaries and pelvic peritoneum, yet endometriotic lesions are also found in other sites such as the abdominal wall, fallopian tubes (FT), bowels, bladder, cervix and vagina [5].

The pathogenesis of endometriosis is still unknown and the most commonly accepted theory proposes endometriosis to arise through retrograde menstruation, whereby shed menstrual endometrial tissue fragments are refluxed into the pelvic cavity, via the FT, and are deposited to form ectopic endometriotic lesions [6,7]. Whilst Sampson’s “transplantation theory” of retrograde menstruation supports the origin of peritoneal lesions, endometriosis occurring outside the female pelvis has been explained by lymphatic or haematogenous spread [8]. In contrast, there are existing theories that refute the concept of endometrial “spreading”; an example of this is the in situ development theory of coelomic metaplasia, whereby endometrial tissue develops from local tissues [9]. There is also ample evidence for the role of endometrial stem cells in endometriosis neogenesis [10,11]. Endometrial stem/progenitor cells have long been proposed as the mechanism by which the endometrium regenerates after menstrual shedding, and several studies have identified putative stem cells in the basalis [10]. It has been proposed that these progenitor cells may be abnormally shed and deposited via retrograde menstruation [12] or be transported to extrauterine sites via the lymphatics or vasculature [13]. In addition, abnormal physiological processes including hormonal dysregulation, inflammation and aberrant immune function are all important contributors to the aetiology of endometriosis [4].

The pathogenesis of endometriosis is evidently complex, as mechanisms of initiation depend on the anatomical distribution of endometriotic lesions. However, amongst the numerous theories and plethora of mechanisms, contributions of the FT to endometriosis are rarely discussed. The tubal endometriosis subtype is sometimes defined collectively as pelvic endometriosis, where the “transplantation theory” can then be applied. However, whilst this theory encompasses reflux of endometrial cells from the oviducts, these structures are rarely affected despite being exposed first [14]. Tubal endometriosis may be associated with tubal ligation, since endometrialisation of the tubal mucosa has been observed following this procedure [15,16]. Inflammation, a hallmark of endometriosis, is induced by tubal endometriosis and may contribute to FT dysfunction and disease progression [17,18]. As well as contributing to other endometriosis subtypes, desquamation of the endosalpinx may play a pivotal role in ovarian endometrioma formation, which are known precursors of malignant neoplasms [19].

This review examines the available evidence for the contribution of the human FT to the pathogenesis, origin, symptoms and clinical consequences of endometriosis. Lastly, we examine the broader topic linking the FT epithelium to the genesis of ovarian epithelial cancers. The presence of endometriotic lesions in the FT and in pelvic/ovarian locations can significantly impact normal tubal functions. Since the diagnosis of tubal endometriosis requires histological examination of the tube, i.e., salpingectomy, we will not be able to decipher the specific influence of tubal endometriosis from the mere functional alterations of the FT that occur when lesions are present at extra-tubal sites. 

## 2. Pathophysiology and Pathogenesis

### 2.1. Common Embryological Origin of the Endometrium and Fallopian Tubes

The FT and uterus share a common developmental origin; they are both derived from the embryonic paramesonephric (Müllerian) ducts. During the 8th and 9th weeks of embryological development, the primary Müllerian ducts differentiate from other gonadal structures, due to the lack of Müllerian-inhibiting substance (MIS), which is secreted only from male sertoli cells. Around the 10th week of development, the two Müllerian ducts fuse to form the uterus, while the non-fused proximal paramesonephric ducts form the FT. The fused medial walls of the ducts are gradually reabsorbed to establish the final morphology of the uterine cavity by the 20th week [20]. The lumen of the FT and uterine cavity therefore exist as an anatomical continuum, and thus, share common features in cell types and progenitors that also exist in endometriotic lesions [21] (Figure 1). However, the distinct hormone responsiveness and different regeneration patterns observed in the two mucosal layers are unanticipated and yet to be fully understood. 

### 2.2. Types of Tubal Endometriosis

Tubal endometriosis can be classified into three distinct histopathological categories (Figure 2a). The first and most common type involves the invasion of endometrial implants into the tubal serosa or subserosa, involving the peritoneal surface of the tubes [22] (Figure 2b). This commonly coexists with other endometriotic lesions within the pelvis, thus can also be included in the broad definition of peritoneal endometriosis. Repeated cycles of haemorrhage result in fibrosis, which can lead to retraction of the tube and may cause hydrosalpinx formation [23] (Figure 2b). Due to its location, the pathogenesis of this tubal endometriosis subtype is expected to be related to the retrograde menstruation theory.

Secondly, endometriotic implants can invade the tubal mucosa, which is known as “endometrial colonisation” and is considered to have a distinct pathogenesis [24]. This can be part of a normal morphological variation; endometrial mucosa has been reported to replace the mucosa of the tubal interstitium in 25% and tubal isthmus in 10% of women [25]. However, ectopic endometrial tissue can grow into the tubal lumen and cause obstruction, which is known as “intraluminal endometriosis”. This can create a vicious cycle, in which the obstruction can cause endometrial tissue within the menstrual flow to be implanted into the proximal end of the tube. This may result in propagation of tubal endometriotic lesions, which will further exacerbate the obstruction [24]. Additionally, cyclical haemorrhage of these implants can cause distension of the FT, leading to haematosalpinx formation [22].

The third type is known as “post-salpingectomy endometriosis”, in which endometriosis develops in the proximal residual segment of the FT, after tubal ligation or salpingectomy. The ectopically placed endometrium may localise in any layer of the tube. This form of tubal endometriosis has been found in 20–50% of tubes examined after ligation, and typically occurs 1 to 4 years following the procedure [26]. An increased incidence is associated with long post tubal ligation intervals, short proximal stumps and the use of electrocautery for ligation [25]. These three forms of tubal endometriosis have distinctly different pathologies and consequently affect different patient groups and may give rise to different symptoms.

Endosalpingiosis is FT epithelial tissue (lacking stroma) located outside the tube, whilst endometriosis is defined as the presence of both endometrial epithelial and stromal-like cells within the same ectopic lesion. Endosalpingiosis could be responsible for the tubal phenotype, whilst the presence of stroma in endometriotic lesions may allow for endometrial differentiation. A potential caveat here is that stromal endometriosis can also occur, with no glandular elements. Papillary tubal hyperplasia has also been proposed as being responsible for endosalpingiosis [27].

Regarding the anatomical distribution of FT endometriosis, lesions of the proximal tube have been shown to mainly affect the mucosa, whilst lesions of the distal tube tend to affect the serosa/subserosa [24]. Some authors have proposed that only lesions beyond the isthmus should be considered as tubal endometriosis, whilst those proximal to the isthmus could be defined as endometrial colonisation [28]. It has been reported that only 10% of cases of tubal endometriosis exhibit lesions in the isthmic and/or ampullary portions of the FT [26]. Thus, the limited evidence suggests lesions may be more prevalent beyond the isthmus and ampulla.

### 2.3. Asymmetric Presentation of Tubal Endometriosis 

There is conflicting evidence regarding the lateral distribution of tubal endometriosis. There are studies reporting a higher prevalence of left-sided tubal endometriosis (4.3–55%) compared with right-sided disease (1.6–31%) [29,30]. The left-sided asymmetry of tubal endometriosis is consistent with the typical left-sided asymmetrical prevalence of ovarian endometrioid cysts (63 vs. 35%) [19,31]. In contrast, there are reports of right laterality of tubal endometriotic implants, with a prevalence of 58.5% in the right compared with 41.5% in the left. This relatively large study reported all other endometriotic lesions, except for ovarian superficial implants and FT endometriosis, to be more frequently located on the left-side [14]. It has been suggested that the left-sided predilection for endometriotic lesions could be due to the presence of the sigmoid colon, which obstructs the flow of menstrual reflux and promotes implantation in the left hemipelvis [14]. Additionally, left-right asymmetries in the pelvis have been proposed to be due to the counter-current of blood flow from the veins to the arteries in the left hemipelvis. This is as a result of the left ovarian vein connecting with the renal vein, as opposed to the inferior vena cava on the right side, and creating greater resistance [30]. The apparent left-sided dominance of tubal endometriosis may support the menstrual reflux theory; however, the available data is too heterogeneous to make conclusions on the asymmetrical preference of tubal endometriosis. It is likely that differences in patient numbers, diagnostic methods and classification systems between studies are responsible for discrepancies in the predominant laterality of tubal endometriosis.

### 2.4. Prevalence of Tubal Endometriosis

Very few studies have assessed the prevalence of endometriosis in the FT and findings differ considerably. Whilst the ovaries are the most frequent site of endometriotic lesion deposition [29], the incidence of tubal endometriosis has been reported to vary from 0.29 to 14.48% [14,22,24,32]. The prevalence of tubal endometriosis may vary with the severity of endometriosis; in women with moderate to severe disease the prevalence of endometriotic lesions within the FT is as high as 60% [33] and women with multiorgan involvement of endometriosis present with tubal endometriosis more frequently when compared with single-organ disease [24]. Taken together, the available data suggests that tubal involvement is a more frequent feature in more severe disease.

### 2.5. Shared Features of Hormone Responsiveness in Endometriotic Lesions and Fallopian Tubes 

The ovarian steroid hormones oestradiol (E2) and progesterone are regulators of the endometrium, acting via their respective cognate receptors: oestrogen receptor (ER) and progesterone receptor (PR). E2 is the key hormone for growth and differentiation of endometrial tissue, binding to either ER subtype α or β. Endometrial epithelium proliferates in response to rising E2 levels in the proliferative phase of the menstrual cycle [34,35]. Progesterone is the dominant hormone of the secretory phase, serving to counteract the effects of E2 by inhibiting endometrial glandular proliferation [36]. 

The FT mucosa is considered to be less hormonally responsive than the neighbouring endometrium and may not exhibit similar cyclical proliferative dynamics [37,38,39]. Previous studies have demonstrated that the FT epithelium expresses PR, ERα, ERβ and androgen receptor (AR) [40,41] (Figure 3). Unlike the endometrium, the FT does not appear to display cyclical variation in steroid receptor expression [38,39]. We have recently identified high AR expression to be the main difference between the FT endosalpinx and endometrial epithelium in premenopausal women [38,39].

Endometriosis is an E2 driven disease [42] and progesterone resistance is proposed to be a pivotal requirement for the establishment and maintenance of endometriotic lesions [43]. Endometriotic tissue has been shown to abnormally express ERβ and PR. The pathological overexpression of ERβ, considered to be the result of deficient methylation of the ERβ promoter, stimulates both inflammation [44,45] and loss of PR expression [46]. Surprisingly, there is very little evidence on the presence of AR in ectopic lesions [47]. To date there have not been attempts to distinguish exact differences in epithelial cell phenotype in all of the three endometrial anatomical subregions (lumen, functionalis and basalis) and the tubal epithelium. Since most treatments available for endometriosis are hormonal, their effectiveness relies on the presence of hormone receptors in the ectopic endometrial lesions. Relative hormone insensitivity of the FT is well-known; therefore, it is tempting to postulate that treatment (hormonal) resistant endometriosis may have a tubal phenotype. Future confirmatory studies should examine the phenotype of endometriotic lesions in the tubes and in other locations compared to normally located tubal or endometrial epithelium. This will decipher the differences in function and hormone sensitivity in different types of endometriotic lesions.

## 3. Tubal Origins of Endometriosis

### 3.1. Stem Cell Origin Theories

The hypothesis that stem cells contribute to the establishment and progression of endometriotic lesions has gained wide acceptance [4,10,11,42,48,49]. However, the available direct evidence to confirm this hypothesis remains limited. The stem cell origin theories of endometriosis propose either (1) retrograde menstruation of normal/abnormal endometrial stem cells, establishing ectopic lesions in extra-uterine locations; or (2) a stem cell of peritoneal or other tissue origin giving rise to endometrial-like ectopic lesions in extra-uterine locations. The well-established Sampson’s theory of the pathogenesis of endometriosis requires endometrial tissue to retrogradely translocate into the pelvic cavity during menstruation [7]. However, retrograde menstruation is a relatively common occurrence (> 90%); [50] whilst the prevalence of endometriosis in the general population remains at approximately 10% [1,2]. Furthermore, the postulated location of endometrial stem cells, the basalis, is not usually shed with menstruation. To address this discrepancy, Leyendecker proposed that women who develop endometriosis may have a more primitive, basalis-like (thus stem cell rich) phenotype in the secretory functionalis of the eutopic endometrium [6]. However, this theory could not be fully substantiated by the authors since there were no stem/progenitor cell markers identified at the time. Recent research has addressed this privation with the identification of multiple stem cell markers in the endometrium, and in vivo lineage tracing has proven the clonal origin of glandular epithelium, thus further cementing the existence of multipotent stem cells in the basalis [51]. Additionally, endometrial epithelial cells with a basalis-like phenotype (SSEA-1+ and SOX9+) are enriched in women with endometriosis [42]. In accordance, these cells have previously been shown to have progenitor properties in vitro [52]. Correspondingly, ectopic lesions are also composed of epithelial cells expressing the same basalis progenitor markers, complementing the baboon model of endometriosis findings [42,52]. Interestingly, establishing endometriosis in a baboon model by transplanting curetted endometrium during spontaneous menstruation caused an increase in the appearance of basalis like SSEA-1 and SOX9 positive epithelial cells in the functionalis of the animals [42]. These observations suggest that endometriosis induces changes in the eutopic endometrium, which may play a role in disease propagation and progression. The baboon model substantiates the retrograde menstruation theory, whilst also proposing that stem/progenitor cells are required to be included in the shed menstrual-effluent for generating ectopic lesions.

The endometrial differentiation of a peritoneal stem cell or other tissue-resident progenitor is plausible, but how such differentiation is preferred by a stem cell in an ectopic niche is not understood. Recent studies have confirmed the previous postulation that endometriotic epithelial cells are of clonal origin whilst stromal cells are not [53]. Collectively, the available data suggests that unipotent endometrial epithelial progenitors and stromal cells originating from multiple progenitors (e.g., endometrial or bone marrow derived), may contribute to ectopically placed endometriotic lesions. The endometrial expression of leucine-rich repeat-containing G-protein-coupled receptor 5 (LGR5) has recently been investigated [54]. LGR5 is a marker of stem cells in various epithelia such as the intestinal mucosa [55], gastric mucosa [56], hair follicles [57] and kidneys [58]. In the endometrium, *LGR5* expression was observed not only in the basalis glands, the postulated location of endometrial stem cells, but also in the luminal epithelium [54]. This luminal population of stem/progenitor cells (*LGR5*++, SSEA-1+, SOX9+) may support maintenance of the luminal epithelium on a daily basis [54]; supporting evidence for this theory is seen in rapid *LGR5+* epithelial cell proliferation observed in many other organs upon tissue damage [59,60]. These cells are more regularly shed than those during monthly menstruation and may have the potential to form endometriotic deposits in the peritoneal cavity as well as in the FT if they are sloughed through. In light of these observations, we proposed a novel hypothesis that there may be more than one stem cell niche in the human endometrial epithelial compartment that contributes to endometrial regeneration [10,54]. Intriguingly, epithelial cells lining the FT constitutively express *LGR5* [54], and future studies will need to explore if the *LGR5+* cells from the tubal or endometrial epithelium contribute to peritoneal endometriosis.

### 3.2. Tubal Stem Cells

Evidence for the existence of stem/progenitor cells in the human FT was first reported in 2012, when a population of self-renewing cells were identified that expressed CD44, EpCAM and integrin α6 [61]. In another study, FT stem cells were cloned and characterised as positive for PAX8 and negative for Foxj1 and PAX2 [62]. More recently, the fimbria of the FT was found to contain a subpopulation of cells expressing stem cell markers LGR5, CD44, SSEA-3, SSEA-4 and ALDH to varying degrees. These cells were capable of generating clonally derived organoids that recapitulated the FT epithelial cell content and architecture [63]. Furthermore, transcript levels of these markers are upregulated by E2 or progesterone treatment in vitro, suggesting hormonal regulation of the stem cell phenotype [64]. Since fluid flows between intrauterine and pelvic compartments, we can postulate that the FT epithelium, which exists as a continuum of the endometrial luminal epithelium, may contribute to the regeneration of the endometrium and also give rise to ectopic endometriotic lesions through damage or shedding. Indeed, many ovarian endometriomas appear to originate from the tubal epithelium rather than the endometrium, since tube-specific markers can be found in ovarian lesions [19]. There are currently no studies reporting the expression of LGR5 in ectopic endometriotic lesions. If present, LGR5*+* ectopic lesions may have originated from the basalis, luminal epithelium or FT epithelium, thus representing a novel multisource hypothesis for the origins of endometriosis (Figure 4).

Recently, an intriguing link has been observed between inflammatory disease and stem cell homeostasis in the FT. It is well established that women with pelvic inflammatory disease are at an increased risk of developing endometriosis [65]. Chronic inflammation of the reproductive tract and pelvic cavity can cause endometriosis by reducing the immune response and enhancing the attachment of shed endometrium at ectopic sites [66]. Whilst inflammation is known to induce cell proliferation, its effects on endometrial and FT cells, specifically stem cells, have not been fully elucidated. Intriguingly, infection of FT organoids with *Chlamydia trachomatis* enhances epithelial cell proliferation and increases stemness via LIF signalling [67]. There is currently no evidence linking chlamydia infection to an increased incidence of endometriosis. However, it is plausible that inflammation-induced epithelial proliferation and enhancement of stemness in the endosalpinx could augment the sloughing off and translocation of mucosal fragments, thus propagating ectopic lesions of FT origin.

The general consensus is that iterate menstruation and continuous seeding of endometrial cells into the pelvis are requirements for the initiation, maintenance and propagation of endometriosis. Despite this, there are no reports suggesting a reduction in the incidence, symptoms or regression of the disease after FT occlusion or excision. It may be naive to expect such an outcome, due to the following reasons: (1) as explained above, tubal occlusion or salpingectomy increases the risk of developing tubal endometriosis; (2) if tubes are occluded but not excised, the remaining stem cell rich fimbrial ends may continue to supply the epithelial component of the endometriotic lesions (the stromal element is postulated to arise from multiple stem cells of bone marrow or peritoneal origin [68,69]); (3) tubal occlusion/excision is usually accompanied with cessation of the use of hormonal contraceptives; therefore, the pre-existing lesions will then be subjected to regular cyclical ovarian hormones and may shed progenitors to the pelvic cavity, initiating further endometriotic lesions. Further studies are therefore needed to ascertain the natural history of endometriosis and the involvement of stem cells. 

## 4. Causes of Tubal Dysfunction in Endometriosis

### 4.1. Inflammation

Local inflammation in response to endometrial tissue at extrauterine sites is a signature element of endometriosis pathology. These events are orchestrated by the upregulation of pro-inflammatory cytokines and activation of immune cells, causing pain, fibrosis and adhesion formation [70] (Figure 5). Since the FT is exposed to the peritoneal fluid, the inflammatory environment initiated by pelvic endometriosis can affect tubal function. The FT exhibits an anti-inflammatory microenvironment during the luteal phase, which facilitates embryo transport and fertilisation [71]. However, patients with pelvic endometriosis are known to have higher numbers of macrophages in the FT, which may originate from the peritoneal cavity [72,73]. Macrophages produce nitric oxide, a known mediator of FT contractility, which can induce an inflammatory response at high concentrations [74]. Nitric oxide causes relaxation of the FT musculature, which may impede peristalsis and thus normal tubal functions [75]. As well as increased macrophage numbers, histological analysis of the FT has shown an increase in inflammatory cells in women with pelvic endometriosis, including monocytes, neutrophils and lymphocytes [72]. Neutrophils are present in the peritoneal fluid of patients with endometriosis and may play a role in lesion establishment and inflammation [76]. Aberrant regulation of lymphocytes (T cells, B cells and natural killer cells) also contributes to endometriosis pathology [77]. However, excluding the study by Matsushima et al., no further studies have been conducted on the role of neutrophils or lymphocytes in the FT of women with endometriosis. Neutrophils of the FT are phenotypically distinct from those of peripheral blood, being less cytotoxic and producing more cytokines [78]. Therefore, endometriosis-induced activation of the innate immune response may be differentially regulated in the FT and thus worthy of future investigation.

Endometriosis-induced inflammation is also linked to oxidative stress through increased levels of reactive oxygen species (ROS) and reduced levels of antioxidants to counteract them. Heightened levels of ROS in the tubal fluid of patients with endometriosis may negatively impact both sperm, oocyte and embryo viability, thus perpetuating inflammation-induced subfertility [79,80]. Salpingitis may also be linked to endometriosis; one study found that 33% of patients with ovarian endometriosis had chronic salpingitis [81]. However, whether ovarian endometriosis was in fact a result of salpingitis in those patients presenting with both diseases was not established. Lastly, prostaglandins E_2_ and F are upregulated in the FT of women with endometriosis, most strikingly in the isthmus and ampulla regions during the proliferative phase [82]. Prostaglandins, specifically prostaglandin E_2_, are known mediators of the inflammatory response in endometriosis [83].

In the tubal endometriosis subtype, the level of local inflammation contained within the FT is anticipated to be higher than that observed in pelvic endometriosis. Elevation of inflammatory and ischemic markers have been observed in a rat model of FT endometriosis. In this model, interstitial FT telocytes implicated in the maintenance of tissue homeostasis were found to decrease in number and display ultrastructural abnormalities when oviductal lesions were present. This was accompanied by enhanced tissue fibrosis, thus linking the endometriosis-induced inflammatory response to tubal damage and resultant subfunction [84]. These findings have not yet been translated to human subjects. In women suffering from tubal endometriosis, local inflammation and fibrotic tissue are more commonly observed when ectopic lesions are present on the serosa compared to the mucosa [24]. Integrated transcriptomic and proteomic analysis of tubal fluid and epithelial cells from patients with tubal endometriosis has identified the acute phase response as uniquely activated, suggesting a disparate pathological mechanism in this subtype [18]. 

The origin of proinflammatory FT fluid in pelvic and tubal endometriosis is uncertain. It may be produced in the tubal lumen or transported from the peritoneal cavity, by the peristaltic movements of the FT, or a combination of both. This proinflammatory tubal fluid can adversely affect the ovarian follicles and endometrial lining, thus affecting fertility. On the other hand, there is evidence that salpingectomy in cases of hydrosalpinges without specifying the causative pathology (whether due to salpingitis, endometriosis, adhesions or blockage) prior to IVF treatment, improves outcome [85]. This suggests that not only does the inflammatory fluid contained in the damaged/dilated FT have the potential to negatively influence the endometrium, but the proximity of the tube to the ovary may have a toxic effect causing adverse reproductive outcomes. Unfortunately, studies examining aspirated inflammatory fluid from the FT report conflicting and inconsistent results to explain the therapeutic benefit of salpingectomy when hydrosalpinx is present [86].

### 4.2. Tubal Obstruction, Adhesions and Hydrosalpinx

Causes of tubal dysfunction in endometriosis may be tubal blockage, adhesion formation or hydrosalpinx. All of these disease states are underpinned by inflammation, can coexist and are often interlinked, making it difficult to distinguish the exact contribution of each cause to the final functional abnormalities in the FT. Up to 30% of women with endometriosis exhibit some form of tubal involvement [22,23]. During the normal menstrual cycle, the oestrogen dominant follicular phase demonstrates an enhanced muscle tone, increased tubal secretions from the mucosa and decreased ciliary activity at the utero-tubal junction [87]. This may lead to stasis of the luminal contents and a functional obstruction at the proximal end of the FT. The reverse happens during the progesterone dominant luteal phase of the cycle, where the functional obstruction is resolved. However, failure of this physiological change will result in a complete anatomical obstruction of the FT. Such persistent obstruction may eventually calcify, ultimately resulting in fibrosis and permanent occlusion of the tube.

Persistent intraluminal obstruction can occur as a consequence of local inflammation in the FT and can exist anywhere along its length. Obstruction at the fimbrial end causes a phimosis like picture and clubbing of the edges of the FT [88], which can lead to fluid stagnation in the lumen, forming a hydrosalpinx or less commonly a haematosalpinx. In the latter, menstrual blood becomes stagnant in the tubes due to retrograde flow from the uterus, or blood accumulates from repeated haemorrhage of intraluminal lesions [89]. Tubal endometriosis is associated with a significantly increased risk of hydrosalpinx or haematosalpinx, as reported in a cross sectional study, with a prevalence of 43% in women with tubal endometriosis [24] and increased risk for women with severe disease [90]. Extraluminal obstruction can occur due to scarring of the serosal surface of the FT, which causes shrinkage of the outer surface and kinking of the lumen. Such scarring may result in women with endometriosis as a consequence of the excisional surgery, pelvic infection and inflammation related to endometriosis [89].

## 5. Consequences of Endometriosis-Associated Tubal Dysfunction

### 5.1. Subfertility

Infertility is “a disease of the reproductive system defined by the failure to achieve a clinical pregnancy after 12 months or more of regular unprotected sexual intercourse” [91]. The aetiology of female subfertility has been attributed to and classified as a variety of ovulatory dysfunctions, tubal defects (tubal infertility) and uterine or peritoneal causes [92]. Concordantly, it has been reported that FT diseases, including tubal obstruction and adhesions, account for 20% of clinical subfertility, whilst uterine and peritoneal causes account for 10% of cases in the UK [92]. However, the recorded prevalence of endometriosis in this particular subset of patients varies. The Human Fertilisation and Embryology Authority (HFEA) reports it to be 6%, yet a number of studies consider it to be significantly higher, ranging between 25 and 71% [93,94]. These same studies also noted that 30–50% of those that had endometriosis were infertile. This estimated prevalence suggests that ectopic endometrium in the FT could be one of the main contributing factors to tubal disease infertility.

The FTs control oocyte transport, sperm storage, fertilization, preimplantation embryonic development and embryo transfer to the uterus. Aberration of these key functions is common in those suffering from endometriosis and can cause infertility. Whilst ectopic endometrium can implant in the tubes, tubal endometriosis is rare compared to other subtypes, e.g., ovarian endometriosis [14], albeit since the diagnosis requires salpingectomy, the true incidence is likely to be higher. However, as previously mentioned, normal tubal function is also impaired by pelvic endometriosis in the absence of tubal lesions that do or do not affect patency. Here, we will highlight several key studies relevant to endometriosis-associated FT dysfunction and resultant subfertility. 

Endometriosis has been shown to affect cilia beat frequency and muscle contractility in the FT. Peritoneal fluid from women with endometriosis significantly reduces ciliary beat frequency in the endosalpinx compared to fluid from healthy patients, most likely through its proinflammatory effects [95]. In a study of 35 women with pelvic endometriosis, it was found that the beat frequency of ciliated cells in the tubal ampulla was significantly decreased compared to healthy controls. Ciliary beat frequency was further decreased in the ampulla and isthmus segments of a subgroup of patients with tubal endometriosis, when compared to the control group and those with pelvic endometriosis only. The same trend was observed when comparing the percentage of ciliated cells in the ampulla and isthmus segments. A decrease in longitudinal muscular contractility and contraction frequency was also observed in the pelvic and tubal endometriosis groups [33]. These changes suggest that the transport function of the FT is deficient in endometriosis, particularly when lesions are present in the tube itself. Reduction in ciliary function and smooth muscle contraction may result in the stagnation of luminal contents and cause a rapid progression to tubal blockage and consequent infertility. A potential mechanism of tubal dysperistalsis in endometriosis is the loss of pacemaker cajal-like type of tubal interstitial cells, which modulate smooth muscle contractility in the FT. A study of 10 women with early stage endometriosis found that cajal-like type of tubal interstitial cells were more sparse and damaged compared to tubes from healthy patients, thus linking endometriosis and aberrant FT motility function at the cellular level [96].

Sperm storage and transport in the FT is also impaired by endometriosis. Tubal dysperistalsis has been observed in patients with endometriosis using hysterosalpingoscintigraphy, which demonstrated a significant reduction in sperm transport capacity [97]. The spermatozoa-binding function of the isthmic mucosa is believed to be important for both sperm transport to the site of fertilisation and fertilisation itself. It has been shown that less sperm bind to this region in women suffering from pelvic endometriosis, demonstrating how impairment of tubal function, specifically the sperm binding capacity of the endosalpinx, can lead to infertility in this pathology [98].

### 5.2. Association with Pelvic Pain

The most common presenting symptom initiating investigations in cases with endometriosis, besides infertility, is CPP, which is defined generally as acyclic pain that is perceived and persistent in the pelvis for more than 3–6 months in duration, strong enough to require medical treatment or impair functional ability and is not related to pregnancy [99]. Up to 33% of women with CPP have confirmed endometriosis upon laparoscopic investigation [100]. Pain in endometriosis is very complex and subjective, and its mechanism is yet to be understood. The wide variation in character of pain and its decorrelation with site and extent of disease raises many hypotheses about the pathogenesis and pathways of endometriosis related pain, which have been discussed extensively [101,102,103]. 

Tubal endometriosis as a source of pain is not described in the literature. The FT represents a complex scenario; due to their shared innervation by autonomic nerve fibers distributed along the ovarian and uterine arteries, they also share some of the pain mechanisms and characteristics of both the uterus and ovaries [104]. In addition to the dual sympathetic and parasympathetic mediation of mucosal function and contractility, the FT have modified Pacinian corpuscles in the ampullary submucosa. These Pacinian corpuscles are highly sensitive mechanoreceptors that respond rapidly to vibration and pressure changes [105]. 

Pain caused by tubal endometriosis can be widely attributed to nociceptive, inflammatory and neuropathic mechanisms similar to other endometriosis related pain. Endometriotic lesions are themselves innervated [106,107,108,109]; however, there are concerns around the correlation between nerve fiber density and degree of pain experienced [110]. It has been suggested that nerve growth alteration is correlated to endometriosis itself rather than endometriosis-induced pain [111]. On the other hand, numerous studies have demonstrated a relationship between high concentrations of nerve growth factor, neuronal markers, nerve growth and pain symptoms [112,113,114]. No studies have explored innervation of endometriotic lesions in the FT.

In hollow viscera, like the FT, serosal surfaces are more sensitive to nociceptive pain and the pain pathway can be triggered in response to muscular spasm, ischemia, haemorrhage, physical compression, inflammation or distension of the organ itself [103]. It has been shown that the density of nerve fibers in the isthmus of damaged FT (hydrosalpinges) is reduced compared to healthy FT [115]. This suggests that pain in hydrosalpinges is probably more nociceptive and inflammatory rather than neuropathic in origin.

Hormone modulation might also have an impact on CPP secondary to endometriosis. The high concentration of E2 and ERs in ectopic lesions is believed to enhance production of inflammatory mediators, like prostaglandin E_2_, indirectly by induction of Cox-2 isoenzyme [116,117]. However, the impact of this in tubal endometriosis is debatable since FT mucosa is less hormonally responsive as discussed earlier. Other inflammatory mediators like interleukins (IL-1, IL-6, IL-8) [118,119,120], cytokines (TNF-α) [121] and activated macrophages (monocyte chemotactic protein 1 [122] were found to be raised in the peritoneal fluid of patients with endometriosis in many studies. These inflammatory mediators can cascade immunologic reactions and trigger neuropathic pain in nerves invaded by endometriotic stromal cells [123], which can still impact the neurogenic complex on the serosal surface of the tube, whether it is produced by the tube itself or not. Once again there is no direct evidence for this, highlighting the need for additional research in this area. 

### 5.3. Ectopic Pregnancy 

The evidence for the association between endometriosis and ectopic pregnancy is scarce. The direct effect of endometriosis on the risk of an ectopic pregnancy is somewhat obscured, since endometriosis, independent of site or stage, may cause subfertility; with 30–68% of subfertile women suffering from endometriosis [124,125]. The rate of ectopic pregnancies in women with endometriosis who achieve a spontaneous pregnancy is unknown. The use of assisted reproduction technology (ART) in women with endometriosis-associated infertility further confuses the available data, since these ART methods may also independently increase the risk of ectopic pregnancies [126].

A recent report suggested an increased risk of ectopic pregnancy in women with endometriosis (RR 1.46, 95% CI 1.19–1.80) [127]. Another meta-analysis also concluded that endometriosis was a common association in women with tubal ectopic pregnancies (OR = 2.16), and it also increases the risk of ectopic pregnancies (OR = 2.66) [128]. However, we would assume the overall incidence of ectopic pregnancy in women with endometriosis to be low, (although increased at the individual level, in women who achieve a pregnancy), due to their background risk of infertility.

Tubal dysfunction related to disruption in normal tubal anatomy and histology is known to be the main reason behind tubal ectopic pregnancies. Consequent changes in the tubal environment, alteration in ciliary beats or muscular contractility may lead to stagnation of the embryo in the tube [129,130]. However, when the resected tubes containing an ectopic pregnancy were examined in a case controlled study, no histological evidence of tubal endometriosis was identified [131]. This may be due to the effect of the hormonal milieu of pregnancy on the endometriotic lesions. Tubal inflammation in women with endometriosis may also increase the risk of ectopic pregnancy [132], but women with salpingitis are at higher risk of developing endometriosis. The authors therefore speculated a relationship between salpingitis, endometriosis and ectopic pregnancy [65]. These postulations are, however, yet to be confirmed in prospective studies.

## 6. Association with Cancer 

### 6.1. Incidence of Ovarian Cancer and Association with Endometriosis

Ovarian cancer is the 7th most common cancer in women and causes more than 150,000 annual deaths worldwide [133]. The estimated lifetime risk of ovarian cancer for women in the UK is around 1 in 52 (~1.9%) [134] and endometriosis is estimated to increase the risk by 27–80% [135,136]. However, given a lifetime risk of 1.9% for the general population, this would result in an estimated lifetime risk of only 2.44–3.46% for women with endometriosis, which is still a low overall risk. 

### 6.2. Endometriosis-Associated Ovarian Cancer (EAOC)

It is known that 90% of ovarian cancers are epithelial ovarian carcinomas (EOC), classified in five main subtypes: 70% high grade serous carcinoma (HGSC); 10% endometrioid carcinoma (EC); 10% clear cell carcinoma (CCC); 3% low grade serous carcinoma; < 5% mucinous carcinoma [137]. Endometriosis has been linked to EOC with greatest prevalence in clear-cell (35.9%) and endometrioid (19%) subtypes; the prevalence of endometriosis is much less in serous (4.5%) and mucinous (1.4%) subtypes [138]. A recent large population-based cohort study [139] demonstrated a significant increase in the rate of ovarian EC and CCC in women with a histological diagnosis of endometriosis at least a year prior to the censoring date (autopsy, bilateral salpingo-oophorectomy or ovarian cancer) with an age-adjusted incidence rate ratio (IRR) of 2.56 (95% CI 1.47–4.47) for EC and 2.29 (95% CI 1.24–4.20) for CCC of the ovary. These rates were much higher when women with a synchronous diagnosis of endometriosis (within one year prior to or six months after the censoring date) were included: age-adjusted IRR 29.06 (95% CI 20.66–40.87) for EC and 21.34 (95% CI 14.01–32.51) for CCC. When the synchronous group was included the results also showed an increased risk for serous and mucinous ovarian cancer with age-adjusted IRR 4.19 (95% CI 3.41–5.15) and 5.87 (95% CI 3.58–9.61), respectively. 

### 6.3. Tubal Origins of Ovarian Cancer

Although ovarian cancers were initially thought to originate within the ovaries, recent studies have suggested a majority originate from the distal FT. Evidence of serous tubal intraepithelial carcinoma (STIC) in specimens from risk-reducing salpingo-oophorectomy and the presence of synchronous STIC lesions in patients with HGSC of the ovary and peritoneum, occasioned a consensus statement in 2016 calling for detailed histopathological assessment of the FT in cases of ovarian/tubal/peritoneal carcinoma using the Sectioning and Extensive Examination of the FIMbrial end (SEE-FIM) protocol. This was to improve accuracy and consistency in primary site assignment, as these lesions are often present in FT with a macroscopically normal appearance [140]. However, since the relationship between STIC lesions and EOC is described in HGSC and since EAOC are mainly CCC and EC, we cannot infer any direct contribution to EAOC from FT.

A recently published study explored the histological diagnosis of endosalpingiosis (ectopic tubal epithelium); they found that 40.4% of cases have concurrent endometriosis and that the increased risk of developing EC and CCC was similar to that of endometriosis even for cases of endosalpingiosis without endometriosis, with an age adjusted IRR of 38.8 (95%CI 29.9–50.4) in the total endosalpingiosis alone group. This risk was reduced to 1.8 (95%CI 0.8–4.0) in the endosalpingiosis group when synchronous cases were excluded [139]. As the probable tubal origin for most EOC has only more recently been recognised, it is likely that the true prevalence of EAOC may be higher than previously reported, as cases of tubal endometriosis (without pelvic endometriosis) would likely be missed without complete SEE-FIM histopathological assessment of the FT.

Given the evidence of tubal origin for the majority of ovarian cancers, FT contributing to ovarian endometriosis and the increased risk of ovarian cancer with endometriosis, attention is now turning to whether the association between endometriosis and ovarian cancer is mediated via the involvement of the FT [19]. Tubal ligation (TL) is known to reduce the risk of EOC by 20–40% [136] especially in EC and CCC, which are associated with endometriosis, supporting the theory that these cancers may have a more proximal tubal or endometrial origin than HGSC, as the distal tube where STIC lesions are found remains in situ after TL. Sterilisation with removal of the fimbrial end of the tube has been shown to be more effective at risk reduction of EOCs than TL, presumably because HGSC is the most common EOC [141].

In summary, the available evidence suggests the FT is likely to contribute to some ovarian endometriotic lesions and possibly ovarian cancers. Further studies tracing the lineage of ovarian endometriotic lesions and tumours to the FT or eutopic endometrium will improve understanding of these interlinked pathologies to find effective preventative and treatment strategies.

## 7. Conclusions

The shared embryological origin and some similar phenotypical and functional features between the FT and endometrium compel us to examine their involvement in the genesis, pathophysiology and clinical consequences of the common yet enigmatic disease, endometriosis. There are many interesting findings suggesting that both endometrium and tubal mucosa may contribute to the origin of the disease and the presence of endometriotic lesions may cause tubal dysfunction that further influences symptomatology. Understanding the differential cellular subtypes constituting both mucosal organs and characterising the various types of endometrial lesions in the context of these cell subtypes will allow us to perceive the origin of the disease as well as more suitable lesion type specific treatments in the future.

## Figures and Tables

**Figure 1 jcm-09-01905-f001:**
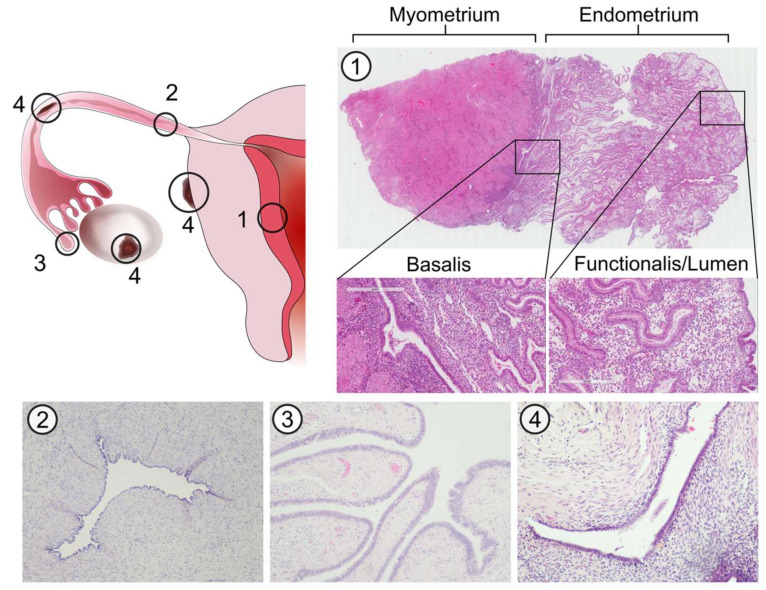
Histological architecture of the endometrium, fallopian tubes (FT) and endometriotic lesions. Micrographs show H&E stained sections of the endometrium (1), transverse planes of the FT isthmus (2) and fimbriae (3) and an endometriotic lesion (4). The endometrium can be divided into three discrete layers based on epithelial cell phenotype; the basalis, functionalis and lumen. Unlike the endometrium, the FT has a single convoluted layer of epithelial cells lining the lumen (endosalpinx) surrounded by stroma. Endometriotic lesions are characterised by the presence of both stromal and epithelial components, the latter of which may exhibit a glandular or luminal phenotype.

**Figure 2 jcm-09-01905-f002:**
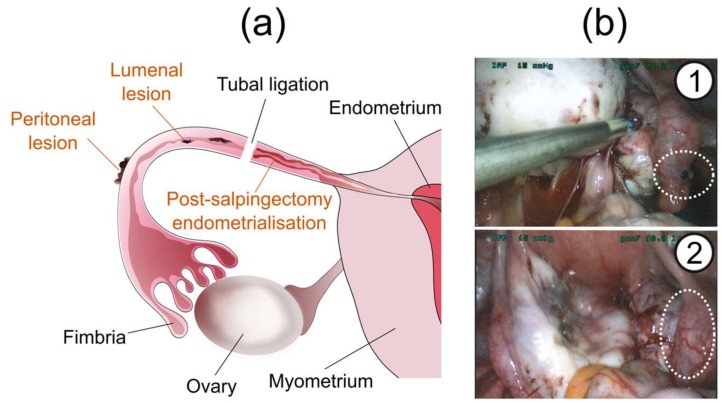
Tubal endometriosis. (**a**) Manifestations of tubal endometriosis. (**b**) Peritoneal endometriotic lesion on the tubal serosa (*1*) and hydrosalpinx resulting from endometriosis-induced tubal occlusion (*2*).

**Figure 3 jcm-09-01905-f003:**
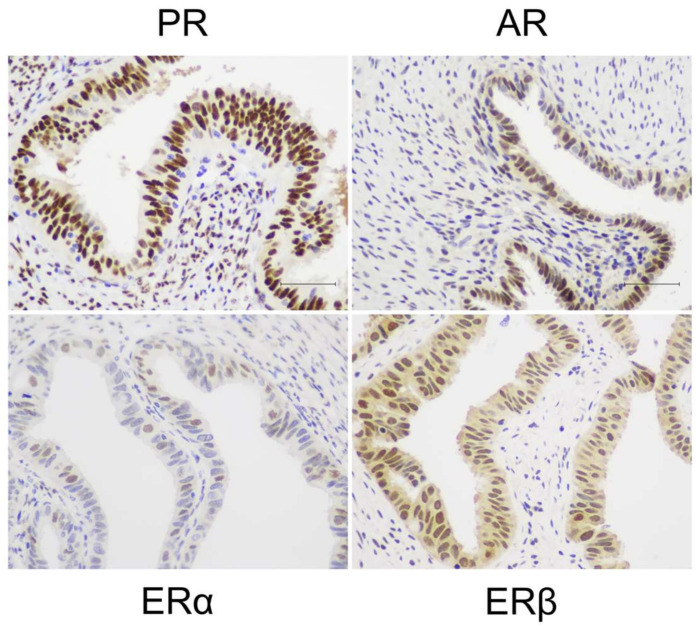
Steroid hormone receptor expression in the FT. Representative micrographs show chromogenic immunostaining of progesterone receptor (PR), androgen receptor (AR), oestrogen receptor α (ERα) and oestrogen receptor β (ERβ) in the fimbriae of the FT.

**Figure 4 jcm-09-01905-f004:**
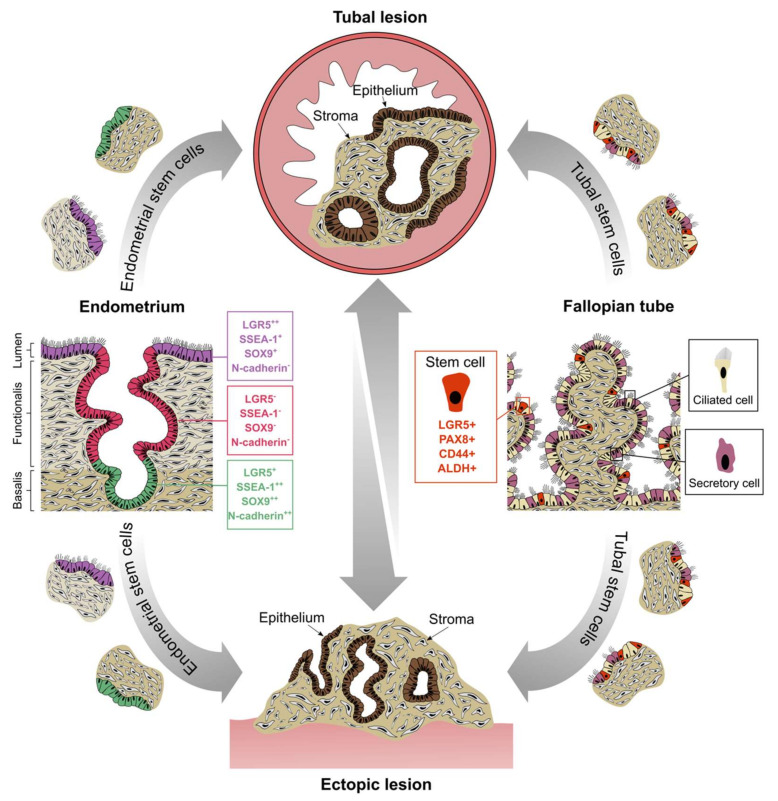
Schematic illustration of the proposed stem cell origins of tubal and pelvic endometriosis. Putative stem cell niches exist in the endometrial basalis (LGR5^+^, SSEA-1^++^, SOX9^++^ and N-cadherin^++^), lumen (LGR5^++^, SSEA-1^+^ and SOX9^+^) and FT epithelium (LGR5^+^, PAX8^+^, CD44^+^ and ALDH^+^). Endometrial fragments containing stem cells may be refluxed into the peritoneal cavity via the tubes, giving rise to both tubal and pelvic endometriosis. Analogously, desquamation of the endosalpinx through injury or natural shedding may also deposit fragments within the tubal lumen and out into the peritoneal space, thus establishing endometriotic explants of tubal origin. Once established, lesions arising from either the endometrium or the FT can spread to secondary sites.

**Figure 5 jcm-09-01905-f005:**
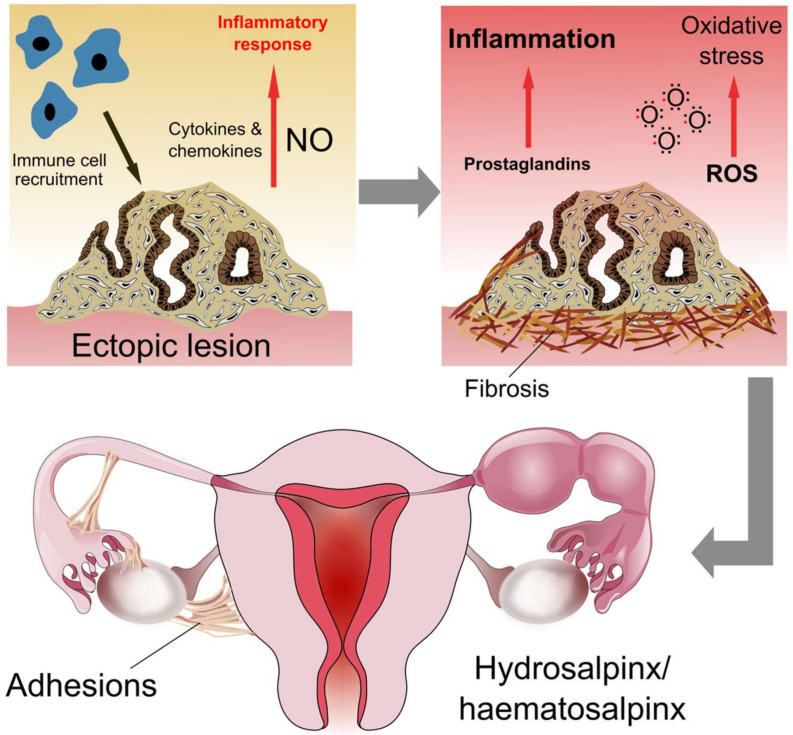
Inflammation in tubal endometriosis. Adhesion and proliferation of endometrial tissue at ectopic sites leads to an influx of immune cells, secretion of cytokines/chemokines and initiation of the inflammatory response pathway. High concentrations of nitric oxide (NO) produced by macrophages promotes the inflammatory environment. Coupled with oxidative stress and prostaglandin-induced maintenance of the inflammatory state, tissue fibrosis is favoured at lesion sites. When lesions are present in the tubal serosa or endosalpinx, fibrosis can result in hydrosalpinx/haematosalpinx and adhesion formation.

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
