# Peer review of "Endometriosis and the Fallopian Tubes: Theories of Origin and Clinical Implications"

_jcm, 2020, doi:10.3390/jcm9061905_

Round 1

Reviewer 1 Report

The review "Endometriosis and The Fallopian Tubes: Theories of Origin and Clinical Implications" describe the commonalities between endometrium and endometriosis lesions and the fallopian tubes, it also describes the different types of tubal endometriosis, hormone response, implications of stem cells in the development of tubal endometriosis, relation with inflammation, subfertility, pain and association with ovarian cancer.

As a whole, it is a very well written manuscript and I only have some minor comments:

  1. Line 47. Authors should specify that transplantation theory refers to Sampson's retrograde menstruation theory.
  2. Figure 1. Specify where the endometriotic lesion is located. It should be specified that it is an ovarian endometriosis lesion.
  3. In line 105, the authors should specify that they are talking about peritoneal endometriosis (it is reflected in the figure but not in the text).
  4. Line 152. What is the prevalence of ovarian endometrioid cysts in the left-side?
  5. Line 154-155. Which other types of endometriosis were included in the study apart from ovarian and FT endometriosis?
  6. Line 187. AR should be in parentheses in line 185, as it has not been previously defined.
  7. Regarding LGR5, the authors explain that endometrial LGR5 cells have potential to form endometriotic deposits in the FT if they are sloughed through the FT. It is described that LGR5 is constitutively expressed in FT and it could have an implication in endometrial regeneration. However, it is not clear how these cells could contribute to endometrial regeneration and which role would they have in tubal endometriosis.
  8. Line 269. There is no evidence that FT epithelium sheds during menstruation. How would FT epithelium contribute to the regeneration of endometrium? By the epithelial secretion through the fluid? Please, clarify.
  9. Line 296. It is stated that the stromal compartment of endometriotic lesions arise from stem cells from bone marrow or peritoneal origin. A reference for this statement is missing.
  10. Line 306. Regarding the figure, it should be LGR5++, SSEA-1++, SOX9++ and N-Cad++. In the text they only have one +.
  11. Line 307. As the previous comment, in the text, LGR5 should be LGR5++, not LGR5+.  
  12. In 4.1. Inflammation,the authors mention that macrophages produce ROS. However, are there other studies that have investigated the role of other immune populations in FT and their role in endometriosis?
  13. Line 334. Are they referring to prostaglandin E2? Or there is overexpression of other prostaglandins E?
  14. Line 377. Which kind of tubal secretions?
  15. Lines 379-383. Is there any evidence that FT cells in endometriosis could be progesterone-resistant that could lead to the failure of resolving the tubal obstruction?

Reviewer 2 Report

This a well-written comprehensive review of the topic of FT endometriosis.  While focusing on the pathophysiology of tubal endometriosis, the authors dived into the association of FT endometriosis with ovarian cancer (point # 6, line 513).  In this context, they started by a general introduction that did not add much to the scientific content and then presented STIC as the precursor of high grade serous ovarian carcinoma.  While this is a well-accepted theory, it is by no means the only explanation of high grade serous carcinoma.  From this position, the authors proceeded to discuss endometriosis-associated ovarian cancer, which is neither high grade serous nor FT-related.  All of this discussion has very little, if anything, to do with FT endometriosis.  

It is recommended to remove point # 6 from this manuscript, adjust the abstract accordingly to remove reference to ovarian cancerogenesis, and to remove unused references.

Reviewer 3 Report

This review is  a nice (but incomplete) review .

The figures are of excellent quality

Nevertheless , there are  3 main concerns 

1) the authors claim in several parts  that there is "growing evidence of ...altough clearly ,it is not fully demonstrated (ex: association with the ovarian cancer ...; also line 577 is confusing  ...)

2) the scientific  content is  sometimes questionnable ...the authors based their discussion on the retrograde menstruation theory  .....but the metaplasia theory   may explain some locations  like  the ovary  where ovarian epithelial inclusions were described in fetal and newborn ovary ( See Motta  et al )

Moreover ,the  different pathways  are not clearly  explained . The influence of ROS ,oxydative  stress , inflammation ,peritoneal mileu ,iron overload  ....are neither scientifically explained  nor well referenced

3)many references  are missing. Tubal endometriosis after different types of tubal sterilization was  described by Donnez  in 1984.
This reviewer is surprised not to see in a review , reference to many important papers 

Minor comments :

1)endometriosis on the  serosa of the tube is not really tubal endometriosis ... just   a site of peritoneal endometriosis 

2)hydrosalpinges are in the high majority of cases the consequence of severe tubo-ovarian adhesions (line  368 . This reviewer disagrees with )

3) How explain controversial data concerning the laterality ...please expain 

4) endosalpingiosis is a  distinct entity. Please ,discuss  the papers reporting histology of the tubal isthmic portion;

Round 2

Reviewer 3 Report

The authors  have answered to all comments of this reviewer